# Development and Biochemical Characterization of Self-Immolative Linker Containing GnRH-III-Drug Conjugates

**DOI:** 10.3390/ijms23095071

**Published:** 2022-05-03

**Authors:** Sabine Schuster, Éva Juhász, Gábor Halmos, Ines Neundorf, Cesare Gennari, Gábor Mező

**Affiliations:** 1Faculty of Science, Institute of Chemistry, Eötvös Loránd University, 1117 Budapest, Hungary; sabine.schuster83@gmail.com; 2ELKH-ELTE Research Group of Peptide Chemistry, Faculty of Science, Eötvös Loránd University, 1117 Budapest, Hungary; 3Department of Pediatrics, Faculty of Medicine, University of Debrecen, 4032 Debrecen, Hungary; juhasze@med.unideb.hu; 4Department of Biopharmacy, Faculty of Pharmacy, University of Debrecen, 4032 Debrecen, Hungary; halmos.gabor@pharm.unideb.hu; 5Department of Chemistry, Institute of Biochemistry, University of Cologne, 50674 Cologne, Germany; ines.neundorf@uni-koeln.de; 6Dipartimento di Chimica, Università degli Studi di Milano, 20133 Milano, Italy; cesare.gennari@unimi.it

**Keywords:** targeted cancer therapy, drug delivery system, gonadotropin releasing hormone, daunorubicin, paclitaxel, peptide-drug conjugates, SMDC, cathepsin B, antitumor activity

## Abstract

The human gonadotropin releasing hormone (GnRH-I) and its sea lamprey analogue GnRH-III specifically bind to GnRH receptors on cancer cells and can be used as targeting moieties for targeted tumor therapy. Considering that the selective release of drugs in cancer cells is of high relevance, we were encouraged to develop cleavable, self-immolative GnRH-III-drug conjugates which consist of a *p*-aminobenzyloxycarbonlyl (PABC) spacer between a cathepsin B-cleavable dipeptide (Val-Ala, Val-Cit) and the classical anticancer drugs daunorubicin (Dau) and paclitaxel (PTX). Alongside these compounds, non-cleavable GnRH-III-drug conjugates were also synthesized, and all compounds were analyzed for their antiproliferative activity. The cleavable GnRH-III bioconjugates revealed a growth inhibitory effect on GnRH receptor-expressing A2780 ovarian cancer cells, while their activity was reduced on Panc-1 pancreatic cancer cells exhibiting a lower GnRH receptor level. Moreover, the antiproliferative activity of the non-cleavable counterparts was strongly reduced. Additionally, the efficient cleavage of the Val-Ala linker and the subsequent release of the drugs could be verified by lysosomal degradation studies, while radioligand binding studies ensured that the GnRH-III-drug conjugates bound to the GnRH receptor with high affinity. Our results underline the high value of GnRH-III-based homing devices and the application of cathepsin B-cleavable linker systems for the development of small molecule drug conjugates (SMDCs).

## 1. Introduction

Drug delivery systems (DDS) are promising tools for targeted tumor therapy providing the selective delivery of cytotoxic drugs to malignant cells, while side-effects and systemic toxicity are reduced. In addition to monoclonal antibodies (mAb), peptide ligands with a high affinity for tumor-specific cell surface compartments (e.g., receptors) can be used as carriers for cytotoxic payloads, as they provide beneficial features such as good tissue penetration, low immunogenicity and structural simplicity which enables their cost-efficient production by chemical synthesis [1,2]. Receptors for the human gonadotropin releasing hormone (GnRH-I, <EHWSYGLRPG-NH_2_, <E is pyroglutamic acid) were not only identified in the pituitary, but also in various reproductive system-related cancers, such as breast, prostate and ovarian cancers, as well as non-reproductive cancers, such as colon and lung cancer [3]. Thus, GnRH-related peptides are promising homing devices to deliver cytotoxic drugs selectively to cancer cells.

A natural isoform of GnRH-I is the sea lamprey analog GnRH-III [4]. This weak GnRH agonist binds to GnRH receptors (GnRH-R) on cancer cells and induces, like GnRH-I, a direct antitumor activity on several cancer cell lines, but its gonadotropin releasing activity is 500–1000 times lower in vitro and in vivo [5,6]. Due to the direct anticancer activity and the low endocrine effect, GnRH-III and its derivatives have been successfully used as homing devices in in vitro and in vivo experiments. In recent years, a large number of GnRH-III-based small molecule drug conjugates (SMDC) have been developed in our laboratories, whereby anthracyclines such as doxorubicin (Dox) or daunorubicin (Dau) were conjugated to the GnRH-III carrier using different linker systems, such as ester, hydrazone, oxime and amide bond [7,8]. In all cases, the lysine in position 8 was used as the ligation site and different spacer moieties have been incorporated at the lysine side chain to conjugate the drug, such as an aminooxyacetic acid (Aoa) linker in the case of the oxime bond formation. Although the oxime linkage does not facilitate the release of the free drug, these SMDCs displayed a substantial in vitro cytostatic effect in combination with the highest chemical and enzymatic stability. Therefore, this linkage system was used for a systematic refinement of the targeting moiety leading to the development of different series of oxime bond-linked GnRH-III-[^8^Lys(Dau=Aoa)] conjugates. Manea et al. reported that an exchange of the ^4^Ser by an acetylated lysine results in an improved cytostatic effect of the conjugate in vitro and in vivo [9]. Additional studies demonstrated that this effect could be further improved using other short chain fatty acids for acylation. The best cytostatic effect was detected for the compound with butyrylated lysine (^4^Lys(Bu)) [10]. On this basis, the influence of further amino acid substitutions within the GnRH-III sequence has been studied with the aim to further improve the antitumor activity of the targeting sequence [11,12]. It could be demonstrated that the in vitro and in vivo antitumor activity was additionally enhanced by combining the deletion of the histidine in position 2, the unnatural amino acid D-1,2,3,4-tetrahydroiso-quinoline-3-carboxylic acid (D-Tic) in position 3 and the butyrylated lysine in position 4 (^2^ΔHis-^3^D-Tic-^4^Lys(Bu)) [12,13]. Furthermore, it has been pointed out that the improved cellular uptake of the conjugate, as well as the fast delivery of the drug to the nuclei have a positive impact on the antiproliferative activity. Moreover, the lysosomal degradation of the compound and the corresponding release of the smallest drug-containing metabolite (H-Lys(Dau=Aoa)-OH) were not substantially affected, thus ensuring the anticancer activity of the conjugate.

Although the usage of non-degradable drug linker systems represents a valuable strategy to deliver drugs efficiently to cancer cells, as shown for the FDA approved antibody drug conjugates (ADCs) trastuzumab emtansine (Kadcyla) and belantamab mafodotin-blmf (Blenrep) [14,15,16], the release of the free drug might enhance the potency of the DDS. Thus, for the majority of the FDA-approved ADCs, the payload is conjugated to the mAb via labile linkers which are relatively stable within circulation, but degradable inside the cancer cell [17,18]. Next to acid labile linkers, enzyme-cleavable peptides have been successfully used in combination with a self-immolative *para*-aminobenzyloxycarbonyl (PABC) spacer. For instance, in the case of four FDA-approved ADCs, the cytotoxic payload monomethyl auristatin E (MMAE) is connected to the mAb by the cathepsin B-cleavable, self-immolative linker system Val-Cit-PABC-MMAE [17,18]. This Val-Cit-PABC linker system was initially applied for the conjugation of Dox to a chimeric mAb resulting in an ADC which revealed excellent stability in human plasma and rapid release of free Dox by cathepsin B and in lysosomal preparation [19]. Other frequently used cathepsin B-cleavable dipeptides are Phe-Lys and Val-Ala, whereby Val-Cit and Val-Ala are the most successfully used cleavable linkers which might be related to the reduced plasma half-life of the Phe-Lys linker [19,20,21].

In comparison to non-cleavable linkers, it could be shown that ADCs with cleavable linker systems reveal not only toxic effects on antigen-positive cancer cells, but also on antigen-negative cancer cells in the direct proximity to these cells. This so called bystander effect is probably caused by direct diffusion of the free drug across the plasma membrane to adjacent cells and/or by extracellular matrix proteins of the microenvironment of the tumor [22,23]. Especially for tumors with a heterogeneous population of antigen-positive and antigen-negative cells, the application of DDSs which exhibit significant bystander killing is assumed to decrease the chances of tumor relapse with a monotherapy [23].

Encouraged by these promising findings, we report on the synthesis and biochemical characterization of eight cleavable self-immolative linker containing GnRH-III-drug conjugates. Of particular interest was the comparison of (1) two GnRH-III targeting moieties (GnRH-III-[^4^Lys(Bu)] (**I**) and GnRH-III-[^2^ΔHis,^3^D-Tic,^4^Lys(Bu)] (**II**)), (2) two cathepsin B-cleavable dipeptidyl-PABC linkers (Val-Ala and Val-Cit) and (3) two traditional anticancer drugs with different modes of action (Dau and PTX). For a better comparison and to demonstrate the proof of concept, four corresponding non-cleavable GnRH-III-Dau and -PTX conjugates have been developed and analyzed. The ^8^Lys of the targeting peptide was used as the ligation site. In the case of the Dau conjugates, the amino group of the daunosamine sugar has been used for attachment to the linker, while in the case of PTX, the C2′-OH group was exploited for this purpose. All synthesized GnRH-III-Dau and -PTX conjugates were studied for their anticancer activity on A2780 ovarian and Panc-1 pancreatic cancer cells. Furthermore, the release of the drug by lysosomal enzymes and the GnRH-R binding affinities of the SMDC were examined.

## 2. Results

### 2.1. Synthesis

#### 2.1.1. Synthesis of Cleavable Linker Containing GnRH-Drug Conjugates

The peptide carriers of the GnRH conjugates **I** and **II** were synthesized by standard Fmoc-SPPS (solid-phase peptide synthesis), while the dipeptidyl-PABC-drug linkers were synthesized in solution as shown in Figure 1 [19,20,24,25]. All intermediates and the final compounds were identified either by NMR or ESI-MS analysis. Initially, the intermediates Fmoc-Val-Cit-OH (**2a**) and Fmoc-Val-Ala-OH (**2b**) were prepared in accordance to Dubowchik et al. and Hochdörffer et al., whereby the obtained yields (**2a**: 77%, **2b**: 64%) were comparable to the reported ones [19,25]. In the next step, 4-aminobenzyl alcohol was coupled affording compound **3a** and **3b** with high yield (<84%). Afterwards, the hydroxyl group was activated by formation of *para*-nitrophenyl carbonate intermediates (**4a**, **4b**). For the synthesis of the Dau-containing linkers, Dau·HCl was coupled to these compounds yielding 47% of Fmoc-Val-Cit-PABC-Dau (**5a**) and 34% of Fmoc-Val-Ala-PABC-Dau (**5b**) after flash chromatography. The yield of **5a** is in line with the reported yield of Z-Val-Cit-PABC-Dox [19]. Then, the Fmoc group was removed in solution and the compounds were used for the coupling with glutaric anhydride without further purification. The obtained products were purified by RP-HPLC which afforded both linkers (**6a, b**) in good yields. The synthesis of the PTX conjugates was continued by the reaction of the derivative **4a** and **4b** with *N*-Boc-*N*,*N′*-dimethylethylene diamine (**18**). After Fmoc deprotection of the compound, both intermediates were treated with glutaric anhydride yielding compounds **8a** and **8b** in adequate yields. The linker synthesis was continued by Boc deprotection and coupling of the activated PTX (**19**) to the diamine moiety. After purification by preparative RP-HPLC, the final drug-linkers **9a** and **9b** were obtained in satisfactory yields and were used for the conjugation to the peptide carriers. To yield the final conjugates, the linkers **6a**, **b** and **9a, b** were preincubated with HATU and the peptide carrier (**I** or **II**) was added. The GnRH-III-Val-Aaa-PABC-drug conjugates (Dau: **10–13**, PTX: **14–17**) were isolated and characterized by analytical RP-HPLC and ESI-MS (Appendix A). All cleavable GnRH-III drug conjugates were obtained in high purity (≥97%) and the conjugation to the targeting moiety proceeded with yields up to 84%.

#### 2.1.2. Synthesis of Non-Cleavable Linker-Containing GnRH-III-Drug Conjugates

The non-cleavable GnRH-III-drug conjugates were prepared as shown in Figure 2. The drug-linkers were coupled to the GnRH-III-carrier as described for the cleavable counterparts, affording the non-cleavable GnRH-III-drug conjugates (**23–26**) in moderate yields (50–72%). Analytical RP-HPLC and ESI-MS analyses evidenced the high purity of the compounds (Appendix A).

### 2.2. In Vitro Cytotoxic Effect

To investigate the anticancer activity of the GnRH-III drug conjugates, cell viability studies have been performed on A2780 ovarian cancer and Panc-1 pancreatic cancer cells. The GnRH-R expression of these cell lines was determined by Western blot studies (Appendix A). In the case of the A2780 cells, a distinct band at 38 kDa could be detected which corresponds to the full-length human GnRH-R. In contrast, the signal intensity of the 38 kDa band was much lower for Panc-1 pancreatic cancer cells being in line with our previous results [13].

Thus, the antiproliferative activity of the GnRH-drug conjugates was studied on high-GnRH-R-expressing A2780 cells and low-GnRH-R-expressing Panc-1 cells. Since the release of free Dau and PTX can be assumed, both drugs were used as controls. The cells were treated for either 24 h (Dau conjugates) or six hours (PTX compounds), followed by additional incubation with fresh growth medium until 72 h after treatment initiation. The obtained results (Table 1) reveal, on the one hand, that the non-cleavable linker-containing conjugates possess a reduced anticancer activity in comparison to the cleavable conjugates and, on the other hand, that the activity of the all GnRH-III-drug conjugates was substantially reduced compared to the free drug.

Moreover, all compounds displayed a lower biological activity on Panc-1 cells than on A2780. In the case of the cleavable GnRH-III-Dau conjugates, the IC_50_ values varied between 2.85–11.18 µM on A2780 cells, whereby the best activity was obtained for compound **13** (2.85 µM) which contained the cathepsin B-cleavage site Val-Ala and the GnRH-III-[^2^ΔHis-^3^D-Tic-^4^Lys(Bu)] peptide carrier.

Apart from that, the IC_50_ values of the cleavable PTX conjugates on A2780 cells are in the same sub-micromolar range and vary between 0.51–0.77 µM, while the activity of these conjugates was approximately 10 times lower on Panc-1 cells (5.03–8.15 µM).

### 2.3. Lysosomal Degradation in Presence of Rat Liver Lysosomal Homogenate

Lysosomal degradation studies were carried out to gain a deeper insight into the mode of action of the self-immolative GnRH-III conjugates. According to the literature, the release of the free drug should occur as described in Figure 3 [19,20]. In the case of the Val-Aaa-PABC-Dau conjugates, the free Dau should be released together with carbon dioxide after enzymatic hydrolysis and 1,6-elimination of the PABC-moiety to 4-methylene-2,5-cyclohexadien-1-imine. Recently, we could demonstrate that the sequence variation within the targeting moiety (His-Trp or D-Tic) just has an impact on the stability of the *N*-terminal region, but not on the degradation of the *C*-terminus and the release of the drug [12]. Therefore, the degradation studies were performed, by way of example, with the ^2^His-^3^Trp-containing self-immolative conjugates **10** and **12**, since they revealed a better solubility in acidic aqueous buffer than the ΔHis-D-Tic compounds. For both compounds, the release of free Dau and the formation of several peptide fragments could be confirmed (Figure 1).

The results indicate that the Val-Cit linker was cleaved slightly faster than the Val-Ala linker, though small amounts of Dau could be detected already after 5 min incubation for both conjugates. In general, both cleavable linkers were successfully proteolyzed by lysosomal enzymes and Dau was released within the first hour. In addition, the degradation of the non-cleavable linker conjugates **23** and **24** were studied, confirming the release of the smallest Dau-containing metabolite H-Lys(Dau-glutaryl)-OH (Figure 1, peaks labeled by #), while free Dau could not be detected.

Furthermore, also the proteolysis of the equivalent PTX-containing GnRH-III conjugates (**14**, **16**, **25**, **26**) in the presence of lysosomal enzymes was studied. The obtained degradation profile of the PTX compounds was comparable to that of the Dau conjugates, whereby both cathepsin-cleavable linker systems were cleaved with the same efficiency (Appendix A). The diamine-PTX fragment could be already detected for both cleavable compounds after 5 min of incubation, while this fragment could not be detected in the case of the non-cleavable linker derivatives even after 24 h of incubation. However, although the diamine-PTX fragment was formed quickly, the release of the free PTX could not be detected within 24 h under the applied in vitro conditions. The relatively high stability of the ethylenediamine carbamate construct and the slow in vitro release of PXT were recently studied in detail [26].

### 2.4. Radioligand Binding Studies

Human pituitary and human prostate cancer tissues have been used to evaluate the binding affinities of the new GnRH-III-drug conjugates to GnRH-R. Therefore, increasing compound concentrations were applied and the displacement of radiolabeled [^125^I]-triptorelin from GnRH-Rs was detected. The obtained results were compared with the binding affinities of the oxime bond-linked GnRH-III-Dau conjugates (**I**, **II**). All compounds bind to the receptors with high affinities in the low nanomolar range, while GnRH unrelated peptides such as somatostatin or bombesin were not able to displace the radio-labelled triptorelin. However, in comparison to the GnRH-III-homing peptide (**I**), the self-immolative linker conjugate exhibited a 3- to 10-times reduced affinity to the GnRH receptors (Table 2). Interestingly, most of the PTX-containing cleavable compounds possessed a slightly higher binding affinity than the corresponding Dau-equivalent, even if the targeting sequence and the cathepsin cleavage site remained the same.

## 3. Discussion

Targeted tumor therapy represents a valuable strategy for the selective and efficient treatment of tumors and their metastases. A promising targeted moiety for SMDC is the sea lamprey peptide hormone GnRH-III. Previous studies have shown that the lysine in position 8 offers a suitable conjugation side for different cytotoxic agents [7,8,9,10]. Furthermore, numerous oxime bond-linked GnRH-III-Dau derivatives with modified peptide sequence have been developed and analyzed to optimize the structure of the targeting peptide [10,11,12]. The high potential of the best candidates (**I**, **II**) as homing device in targeted tumor therapy could be confirmed in in vitro studies as well as in vivo on tumor-bearing mice [10,12,13,27].

Although non-cleavable linkers such as oximes provide a higher stability during circulation and show reduced off-target toxicity, 9 of 11 FDA-granted ADCs (status as of December 2021) comprise cleavable linker systems [18]. The main disadvantage of non-cleavable linkers is that the release of the payload depends on the proteolysis of the entire homing device. This is commonly accompanied with the remainder of the amino acid residues on the released payloads which can have an influence on the activity and the cell permeability [18]. The latter is important to elicit bystander effects, especially for the treatment of heterogeneous tumors [22,23].

Considering that for the oxime bond-linked GnRH-III conjugates the smallest Dau-containing metabolite which is released by lysosomal enzymes possesses a lower cytotoxic potential than the free drug [28], the usage of cleavable linker systems might be favorable. Therefore, our aim was the development and evaluation of novel GnRH-III drug conjugates containing the cathepsin B-cleavable dipeptidyl linker Val-Ala or Val-Cit, and the self-immolative PABC moiety. Considering the favorable results of the oxime-bond linked conjugates [12,13], the corresponding peptide sequences (**I**, **II**) have been selected as targeting moieties for the novel conjugates and the classical anticancer drugs Dau and PTX were used as payloads.

Both cathepsin B cleavage motifs have been frequently used for ADCs which are already FDA approved or in (pre)clinical trials [17], since they provide a high stability during circulation but are rapidly cleaved by lysosomal proteases. To promote the accessibility of the cleavage site adjacent to bulky payloads, additional spacers such as PABC are often incorporated [17,20,29]. These spacers can undergo irreversible disassembly directly after cleavage by specific intracellular mechanisms which results in the release of the free drug. Beyond that, a variety of PTX prodrugs have been designed to ligate PTX to different targeting moieties, and to improve the solubility of PTX [20,24,29,30,31,32]. Furthermore, it could be shown that an elongated linker between the cathepsin B cleavage site and PTX might be favorable for a rapid enzymatic cleavage of the dipeptide linker [24]. Therefore, the well-known *N*,*N′*-dimethylethylene diamine spacer was incorporated between the PABC moiety and the PTX [20,24,33].

The Val-Aaa-PABC-containing GnRH-III-drug conjugates (where Aaa is Ala or Cit) were prepared in accordance with the literature [19,20,24,25]. The syntheses of the final linkers and the corresponding intermediates were confirmed by NMR and ESI-MS, and the yields of the compounds were comparable to the literature. To ligate the linker and the targeting moiety, an amide bond formation was carried out. Therefore, the glutaryl-linker was preincubated with a coupling agent before the peptide carrier was added. The formation of the final conjugates could be confirmed by ESI-MS. All compounds were obtained in satisfactory yields and high purities after RP-HPLC purification.

After synthesis, the conjugates were analyzed for their antiproliferative activity on A2780 ovarian cancer and Panc-1 pancreatic cancer cells. The GnRH-R expression of these cell lines was determined by Western blot studies revealing that A2780 cells possessed a high GnRH-R protein level, while in Panc-1 cells the receptor level was substantially reduced which is consistent with previous results [13].

All cleavable GnRH-III-Dau conjugates revealed a significant antiproliferative activity on A2780 cells with IC_50_ values in the low-micromolar range, whereas the activity of the non-cleavable conjugates was strongly decreased. A similar effect could be observed in recent studies which demonstrated that the in vitro antitumor activity of GnRH-III conjugates was remarkably decreased when the amino group of the daunosamine sugar was used for amide bond formation to a glutaryl-spacer [7]. This reduced activity might be related to the fact that only the Dau-containing metabolite H-Lys(Dau-glutaryl)-OH is released by lysosomal enzymes but not the free drug. It can be assumed that the stable acylation at the amino group of the daunosamine sugar moiety prevents an efficient intercalation of the Dau-derivative in the minor groove of DNA [34,35,36,37]. In contrast, the smallest Dau-containing metabolite of the oxime-bond linked Dau conjugates (H-Lys(Dau=Aoa)-OH), on which lysine is connected to the C13 carbonyl group of Dau instead of the amino group, can intercalate in DNA and thereby inhibit topoisomerase II activity, leading to a substantial reduction of cell proliferation [28]. This illustrates that the design of the non-cleavable linker can cause a huge difference of the biological activity.

A comparison of the IC_50_ values of the cleavable GnRH-III-Dau conjugates on A2780 cells emphasizes that the optimized targeting moiety GnRH-III-[^2^ΔHis-^3^D-Tic-^4^Lys(Bu)] had a positive effect on the cytotoxic activity which is in line with our previous results [12,13]. In addition, the anticancer activity of the Val-Ala conjugates (**12**,**13**) was marginally improved compared to the Val-Cit compounds (**10**,**11**), while in the case of the lysosomal cleavage studies, slightly better results could be achieved for the Val-Cit compounds. Both linkers were rapidly cleaved by lysosomal enzymes and free Dau was released already within one hour which is of high relevance for the biological activity of the compounds. If we compare this result to the release of the smallest Dau-containing metabolite of the non-cleavable oxime bond-linked GnRH-III-Dau conjugates [11,12], we can conclude that the release is faster and more efficiently for the conjugates with the cathepsin B-cleavable linker. This might be mainly related to the fact that for cleavable conjugates only one peptide bond needs to be cleaved to release the free drug, while for the non-cleavable conjugates the digestion of the targeting moiety is required to release the Dau-containing metabolite. We recently demonstrated that different lysosomal enzymes might be involved in the release of the metabolite H-Lys(Dau = Aoa)-OH [11]. Similar observations have been made for the degradation of the non-cleavable compounds (**23**,**24**) and the accompanied formation of the metabolite H-Lys(Dau-glutaryl)-OH.

Apart from that, the receptor binding studies also provided better results for the Val-Cit compounds. Thus, the receptor binding affinity of the Val-Cit linker containing compound **11** was twice as high as that of the corresponding Val-Ala conjugate **13**. However, considering that for both homing peptides the IC_50_ values of the Val-Ala and the Val-Cit conjugates varied only by a factor of 1.5, this observation should not be overemphasized.

Beyond that, the radioligand binding studies pointed out that all GnRH-III-drug conjugates bind efficiently and with high affinity to the GnRH-receptors, but the binding affinities of the cleavable GnRH-III DDS were noticeably reduced compared to homing peptide **I**. This effect could not be observed for the recently reported oxime bond-linked GnRH-III-Dau conjugates which showed affinities similar to those of the drug-free GnRH-III peptide **I** [12]. Nonetheless, the obtained IC_50_ values were in the low nanomolar range demonstrating that the radiolabeled GnRH-I agonist [^125^I]-triptorelin was efficiently replaced by the GnRH-III compounds using concentrations between 1 pM and 1 µM. In contrast, unrelated peptides such as somatostatin or bombesin were unable to affect the GnRH-R binding of the radiotracer at concentrations up to 1 µM [38,39]. Therefore, it can be assumed that all cleavable compounds bind specifically and with high affinity to the GnRH-receptor which is important to ensure the receptor-mediated uptake of the GnRH-III-drug derivatives.

In comparison to the GnRH-III-Dau conjugates, all cleavable PTX compounds displayed nearly the same activity on A2780 cells. It can be assumed that this is caused by the PTX-releasing process. Former studies pointed out that in the first step only the PTX pro-drug is released which still contains the diamine linker. The subsequent formation of the 1,3-dimethyl-2-imidazolidinone and the accompanied release of PTX is considered to be the rate limiting of this self-immolative releasing mechanism [24]. The results of the lysosomal degradation studies support this evidence. The cleavage of the Val-Aaa linkers and the corresponding formation of the diamine-PTX metabolite were already detected after 5 min of incubation, whereas the release of free PTX could not be confirmed within 24 h of incubation. These findings might serve as an explanation for the similar IC_50_ values of the PTX conjugates and the wide disparity of the biological activity between the free PTX and the conjugates, since the exposure of the free C2′-OH group is highly important for the activity of PTX [40]. Apart from that, it can be assumed that the acidic conditions of the experiment, which were needed to ensure the activity of the lysosomal enzymes, prevent the cyclisation of the diamine-linker and the release of the free PTX. Considering that the intracellular pH of cancer cells is defined to be ≥7.4 [41], the nucleophilic attack of the secondary amine towards the carbamate function, followed by the formation of the cyclic urea derivative and the subsequent release of the free PTX (Figure 3), might be much more favorable in the cytosol than in lysosomes. This assumption is supported by recent findings from Dal Corso et al. revealing that no significant drug release could be detected at acidic pH (5.5) for an identical diamine spacer, while the cyclisation takes place at pH 7.5 (t_1/2_ = 5.4 h) [26]. In summary, the obtained results are in line with the literature and it could be shown that the cleavage mechanism by lysosomal enzymes and the release of the PTX prodrug occurs efficiently, while the formation of the free PTX is the rate limiting step [20,24,26]. Therefore, it is most likely that this has a stronger effect on the antitumor activity than the targeting sequence or the Val-Aaa spacer. In addition, three of four cleavable PTX conjugates displayed a higher GnRH-R binding affinity than their Dau equivalents. A possible explanation for this observation could be the incorporation of the diamine spacer which might result in a higher flexibility and provide a longer, more favorable distance between the drug and the targeting moiety. Moreover, the inherent properties of the drug in combination with the linker system might also have an impact on the receptor affinity.

In contrast to the activity on A2780 ovarian cancer cells, all applied conjugates revealed substantially decreased cytostatic effect on Panc-1 pancreatic cancer cells. The reduced activity of the conjugates is in correlation with a lower GnRH-R expression level. However, the free drugs Dau and PTX also revealed a decreased anticancer activity on Panc-1 cells. This reduced potency might be related to the chemoresistance of the Panc-1 cells, since it is well known that pancreatic cancers commonly possess several cellular mechanisms which lead to strong resistance towards a variety of classical anticancer drugs [42,43,44]. Apart from that, the anticancer activity of Dau on Panc-1 cells was approximately 10 times lower than on A2780, while the activity of the SMDCs with targeting moiety **II** (**11**,**13**) was more than 20 times lower on Panc-1 cells which might be caused by the reduced expression level of GnRH receptors on Panc-1 cells.

In summary, we demonstrated that cathepsin B-cleavable self-immolative linker systems which are frequently used for ADCs, could be successfully applied to our GnRH-III-based DDSs (Figure 4). The cleavable GnRH-III compounds inhibited the cell proliferation of GnRH-R-expressing cancer cells in a dose-dependent manner, while the activity of the compounds was reduced on cancer cells which possessed a lower GnRH-R expression level. Moreover, all non-cleavable linker-containing GnRH-III-drug conjugates exhibited a clearly decreased growth inhibitory effect on both cell lines compared to the cleavable conjugates. Next to this, the release of free Dau and the PTX prodrug was confirmed by lysosomal degradation studies, providing the proof of concept. Nevertheless, considering that the binding affinities of the novel conjugates were clearly reduced compared to the targeting moiety alone, further optimization of the linker system might be beneficial. Additionally, the application of more potent drugs should also be taken into account. Nonetheless, the present data demonstrate the high value of GnRH-III-based targeting moieties and the promising characteristics of lysosomal cleavable-linker systems for the development of SMDC.

## 4. Materials and Methods

The general information about the used chemical reagents and methods (analytical and preparative RP-HPLC, mass spectrometry and NMR) can be found in the Appendix A.

### 4.1. Synthesis

#### 4.1.1. Synthesis of Peptide Carriers

The synthesis and purification of the GnRH-III-based targeting moieties (**I**,**II**) was carried out as stated recently [12]. After solid phase peptide synthesis and subsequent cleavage from resin, the peptides were isolated by preparative RP-HPLC using water with 0.1% trifluoroacetic acid (TFA) as eluent A and acetonitrile-water 80:20 (*v*/*v*) with 0.1% TFA as eluent B.

#### 4.1.2. Synthesis of Self-Immolative and Non-Cleavable Drug Linker

Fmoc-Val-Cit-PAB-Pnp (**4a**) and Fmoc-Val-Ala-Pnp (**4b**) were synthesized in solution as previously described [19,25]. The corresponding ^1^[H]-NMR data of the compounds and intermediates are given in the Appendix A.

**Fmoc-Val-Cit-PABC-Dau (5a):** Dau (38 mg, 0.072 mmol, 1.1 eq) was dissolved in 1.5 mL dry DMF. Compound **4a** (50 mg, 0.0652 mmol, 1 eq) and DIPEA (17.1 µL, 0.0978 mmol, 1.5 eq) were added and the mixture was stirred until the next day (RT, N_2_ atmosphere). After addition of 70 mL EtOAc, 1 M KHSO_4_ (2 × 10 mL), sat. NaHCO_3_ (2 × 10 mL) and 10 mL brine were used for extraction. Then, Na_2_SO_4_ was used as drying agent and added to the organic layer. After evaporation, the remaining red solid was dissolved in DCM:MeOH(9:1, *v*/*v*), filtered and purified by flash-chromatography (eluents: initially 6:3:1, then 7:2:1 EtOAc/hexane/MeOH, followed by 9:1 DCM/MeOH). Combined product-containing fractions were evaporated affording **5a** as a red solid (47%). (ESI-MS: MW_cal_: 1155.21, found [M+H]^+^ = 1155.63, [M+Na]^+^ = 1177.53, [M-H]^−^ = 1153.31).

**Glutaryl-Val-Cit-PABC-Dau (6a):** Fmoc-group of **5a** (35 mg, 0.0303 mmol) was removed by adding 2 mL DMF and 5 eq piperidine. The solution was stirred for 2 h at RT. DMF was evaporated under high vacuum. The remains were triturated with Et_2_O. Then, the precipitate was isolated by centrifugation and resolved in dry DMF (2 mL). Glutaric anhydride (6.91 mg, 0.0606 mmol, 2 eq) was dissolved in dry DMF (106 µL) and added, followed by addition of DIPEA (10.6 µL, 0.0606 mmol, 2 eq). After 2 h stirring at RT, DMF was removed in vacuo and ACN/water was used to dissolve the remaining red solid for purification by semipreparative RP-HPLC. The **6a**-containing fractions were pooled and freeze-dried (38%). (ESI-MS: MW_cal_: 1047.07, found [M+Na]^+^ = 1069.50, [M-H]^−^ = 1045.72).

**Fmoc-Val-Ala-PABC-Dau (5b):** Dau (32.4 mg, 0.0615 mmol, 1 eq) was dissolved in 2 mL dry DMF. Then compound **4b** (62.8 mg, 0.0922 mmol, 1.5 eq) and DIPEA (20 µL, 0.115 mmol, 1.9 eq) were added. The reaction was stirred overnight at RT under N_2_ atmosphere. Afterwards, 70 mL EtOAc were added and organic phase was extracted with 1 M KHSO_4_ (2 × 10 mL), sat. NaHCO_3_ (2 × 10 mL) and 10 mL brine. The organic layer was dried with Na_2_SO_4_ and concentrated. The red solid was dissolved in DCM:MeOH (9:1, *v*/*v*), filtered and purified by flash-chromatography (eluents: 7:2:1 EtOAc/hexane/MeOH, followed by 9:1 DCM/MeOH). Combined product-containing fractions were evaporated affording Fmoc-Val-Ala-PABC-Dau (**5b**) as a red solid (34%).(ESI-MS: MW_cal_: 1069.11, found [M+Na]^+^ = 1092.04, [M-H]^−^ = 1068.22).

**Glutaryl-Val-Ala-PABC-Dau (6b):** Fmoc-group of **5b** (22.2 mg, 0.0211 mmol) was removed as described for **6a**. The resulting H-Val-Ala-PABC-Dau was resolved in dry DMF (1.5 mL). Glutaric anhydride (64.8 mg, 0.0422 mmol, 2 eq) was dissolved in dry DMF (68 µL) and added, followed by addition of DIPEA (7.4 µL, 0.0422 mmol, 2 eq). The reaction was stirred overnight at RT, then DMF was evaporated under high vacuum and the remaining red solid was dissolved in ACN/water and purified by semipreparative RP-HPLC. Product-containing fractions were combined and freeze dried (65%). (ESI-MS: MW_cal_: 960.97, found, [M-H]^−^ = 959.45).

***N*-(Boc)-*N*,*N′*-dimethylethylenediamine (18):***N*,*N′*-dimethylethylenediamine (1.5 g ≙ 1.86 mL (ρ = 0.807 g/mL), 17.02 mmol, 3 eq) was dissolved under N_2_ atmosphere in dry DCM (20 mL) and cooled to 0 °C. Boc_2_O was dissolved in dry DCM (10 mL) and added very slowly. The reaction was stirred overnight. The solvent was evaporated and 100 mL EtOAc were added. The organic phase was washed with water (2 × 20 mL) and brine (2 × 20 mL), dried with Na_2_SO_4_. The filtrate was concentrated in vacuo offering **18** as a pale yellow oil. ^1^H-NMR (400 MHz, CDCl_3_): δ = 3.36 (bs, 2H), 2.90 (s, 3H), 2.77 (bs, 1H), 2.49 (s, 3H), 1.87 (bs, 2H), 1.48 (s, 9H).

***N*-(Boc)-*N′*-(H-Val-Cit-PABC)-*N*,*N′*-dimethylethylenediamine (7a):** Compound **4a** (250 mg, 0.326 mmol, 1 eq) was dissolved in 1.5 mL dry DMF under N_2_ atmosphere. The solution was cooled to 0 °C and compound **18** (solution of 159 mg in 1 mL DMF, 0.815 mmol, 2.5 eq) was added, followed by addition of DIPEA (142 µL, 0.815 mmol, 2.5 eq). Reaction was warmed up slowly and stirred overnight at RT. EtOAc (100 mL) was added and washed with 0.5 M KHSO_4_ (2 × 30 mL), sat. NaHCO_3_ (2 × 20 mL) and brine (1 × 30 mL). The KHSO_4_ phase was basified with 10 M NaOH to pH 9 and extracted with EtOAc (2 × 50 mL). The organic layer was washed with brine and dried with Na_2_SO_4_ and concentrated (yellow oil, 53%). (ESI-MS: MW_cal_: 593.72, found [M+H]^+^ = 595.33, [M+Na]^+^ = 617.45, [M-H]^−^ = 592.26).

***N*-(Boc)-*N′*-(glutaryl-Val-Cit-PABC)-*N*,*N′*-dimethylethylenediamine (8a):** Compound **7a** (107 mg, 0.1802 mmol, 1 eq) was dissolved in 2 mL DMF. Glutaric anhydride (61.7 mg, 0.541 mmol, 3 eq) was added, followed by addition of DIPEA (95 µL, 0.541 mmol, 3 eq). The reaction was stirred at RT for 4 h. The solution was concentrated under high vacuum and EtOAc (100 mL) was added. The organic phase was washed with 1 M KHSO_4_ (2 × 20 mL) and brine (2 × 10 mL), dried with Na_2_SO_4_ and evaporated. The remained solid was dissolved in 2 mL DCM/MeOH (9:1, *v*/*v*) and purified by flash chromatography (eluent was stepwise changed from 95:5 to 80:20 DCM/MeOH + 0.1% AcOH). Combined fractions were concentrated in vacuo. The oily product was purified by semipreparative RP-HPLC to remove AcOH. The **8a**-containing fractions were combined and freeze dried (37%). (ESI-MS: MW_cal_: 707.81, found [M+Na]^+^ = 730.84, [M-H]^−^ = 706.77).

***N*-[carbonyl-(2′-PTX)]-*N′*-(glutaryl-Val-Cit-PABC)-*N*,*N′*-dimethylethylenediamine (9a):** The Boc-group of compound **8a** (12 mg, 0.0169 mmol) was cleaved in 1.5 mL DCM/TFA (2:1, *v*/*v*) and stirred for 45 min, followed by evaporation. The product was purified by RP-HPLC and lyophilized. The obtained *N*-(glutaryl-Val-Cit-PABC)-*N*,*N′*-dimethylethylenediamine (9.2 mg, 0.01513 mmol, 1 eq) was dissolved in dry DMF. Then, activated PTX **19** [20] (19.8 mg, 0.01943 mmol, 1.3 eq) was added, followed by addition of DIPEA (10.5 µL, 0.06058 mmol, 4 eq). The reaction was stirred for 24 h at RT and purified by preparative RP-HPLC (57%). (ESI-MS: MW_cal_: 1487.60, found [M+H]^+^ = 1487.93).

***N*-(Boc)-*N′*-(Fmoc-Val-Ala-PABC)-*N*,*N′*-dimethylethylenediamine (7b)*:*** Compound **4b** (150 mg, 0.2204 mmol, 1 eq) was dissolved in 10 mL THF under N_2_ atmosphere. The solution was cooled to 0 °C and **18** (solution of 104 mg in 2 mL THF, 0.551 mmol, 2.5 eq) was added, followed by addition of DIPEA (96 µL, 0.551 mmol, 2.5 eq). The reaction was warmed up slowly and stirred overnight at RT. EtOAc (70 mL) was added and washed with 1 M KHSO_4_ (2 × 30 mL), sat. NaHCO_3_ (4 × 15 mL) and brine (1 × 30 mL). The organic layer was dried with Na_2_SO_4_, concentrated and purified by flash chromatography (eluent was stepwise changed from 100% DCM to 4% MeOH in DCM). Product-containing fractions were combined, and evaporation provided **7b** as a yellow solid (66%). (ESI-MS: MW_cal_: 729.86, found [M+Na]^+^ = 752.89).

***N*-(Boc)-*N′*-(glutaryl-Val-Ala-PABC)-*N*,*N′*-dimethylethylenediamine (8b):** Fmoc group of **7b** (60.6 mg, 0.0830 mmol) was deprotected (see **6a**) and resolved in 2 mL DMF. Glutaric anhydride (18.9 mg, 0.1660 mmol, 2 eq) was added, followed by addition of DIPEA (29 µL, 0.1661 mmol, 2 eq) and the reaction was stirred overnight at RT. The solution was concentrated under high vacuum and EtOAc (35 mL) was added. The organic phase was washed with 1 M KHSO_4_ (2 × 7 mL) and brine (1 × 10 mL), dried with Na_2_SO_4_ and evaporated. The remained solid was dissolved in 2 mL DCM/MeOH (9:1, *v*/*v*) and purified by flash chromatography (eluent was stepwise changed from 100% DCM to 10% MeOH in DCM + 0.1% AcOH). Combined fractions were concentrated in vacuo and product-AcOH mixture remained which could be separated by semipreparative RP-HPLC. Compound **8b**-containing fractions were combined and freeze dried (47%). (ESI-MS: MW_cal_: 621.72, found [M+Na]^+^ = 645.29, [M-H]^−^ = 620.74).

***N*-[carbonyl-(2′-PTX)]-*N′*-(glutaryl-Val-Ala-PABC)-*N*,*N′*-dimethylethylenediamine (9b):** Boc-group of **8b** (12 mg, 0.0183 mmol) was cleaved (see **9a**) and the product was purified by semipreparative RP-HPLC and lyophilized. The obtained *N*-(glutaryl-Val-Ala-PABC)-*N*,*N′*-dimethylethylene diamine (4 mg product, 0.0077 mmol, 1 eq) was dissolved in dry DMF and activated PTX (**19**) (12.3 mg, 0.01207 mmol, 1.6 eq) was added, followed by addition of DIPEA (5.3 µL, 0.0307 mmol, 4 eq). The reaction was stirred for 24 h at RT and purified by preparative RP-HPLC (70%). (ESI-MS: MW_cal_: 1401.51, found [M+H]^+^ = 1401.85, [M-H]^−^ = 1400.91).

**Glutaryl-Dau (20):** Dau (25.33 mg, 0.22 µmol) was dissolved in 2 mL dry DMF, glutaric anhydride and DIPEA were added and stirred at RT for 3 h. The mixture was acidified with TFA and linker **20** was purified by preparative RP-HPLC (ESI-MS: MW_cal_: 641.62, found [M+H]^+^ = 642.08, [M-H]^−^ = 640.25, [M-TFA]^−^ = 754.37).

***N*-Boc-*N′*-(glutaryl)-*N*,*N′*-dimethylethylenediamine linker (21):** Intermediate **18** (198 mg, 1.05 mmol, 1.5 eq) was dissolved in 1.5 mL dry DMF. Glutaric anhydride (80 mg, 0.70 mmol, 1 eq) and DIPEA (240 µL, 1.4 mmol, 2 eq) were added and stirred at RT for 6 h. Then EtOAc was added (50 mL) and was washed with 1 M KHSO_4_ (4 × 10 mL) and brine (2 × 10 mL). The organic layer was dried with Na_2_SO_4_ and concentrated by evaporation. Compound **21** was used for the next step without further purification (ESI-MS: MW_cal_: 302.37, found [M+H]^+^ = 303.27, [M-H]^−^ = 301.19).

***N*-[carbonyl-(2′-PTX)]-*N′*-(glutaryl)-*N*,*N′*-dimethylethylenediamine linker (22):** Boc group of compound **21** (100 mg, 1.031 mmol) was removed (see **9a**) and *N*-(glutaryl)-*N*,*N′*-dimethylethylenediamine was obtained as a dark-brown oil which was directly used (1.6 mg, 0.0079 mmol, 1.15 eq) and dissolved in dry DMF (0.5 mL). Activated PTX (**19**) (7 mg, 0.0069 mmol 1 eq) was also dissolved in dry DMF (0.5 mL) added, followed by addition of DIPEA (36 µL, 0.207 mmol, 30 eq, pH 8–9). The reaction mixture was stirred overnight at RT. Mixture was acidified with TFA and linker **22** was purified by semipreparative RP-HPLC (ESI-MS: MWcal: 1082.15, found [M+H]^+^ = 1082.68, [M-H]^−^ = 1081.92, [M+TFA]^−^ = 1194.84)

#### 4.1.3. Conjugation Reaction of Drug-Linker and GnRH-III Peptide

In general, the drug-linker (1 eq) was dissolved in 1 mL dry DMF. HATU (0.9 eq) and DIPEA (2 eq) were added and stirred for 30 min. Then, the peptide carrier (1 eq) was added and stirred overnight at RT. DMF was evaporated and the final conjugates were purified by semipreparative RP-HPLC. The appropriate amount of starting material and the obtained yield of conjugation product are summarized in Appendix A.

### 4.2. In Vitro Antiproliferative Activity

For cultivation of A2780 human ovarian and Panc-1 human pancreatic cancer cells, RPMI-1640 medium with 10% FBS, L-glutamine and 1% Penicillin-Streptomycin were used. Cells were cultured in appropriate plastic flasks at 37 °C (humidified atmosphere, 5% CO_2_/95% air).

A set-up of 96-well plates was used to study the in vitro antiproliferative activity. Therefore, 5 × 10^3^ cells in 100 µL complete medium were seeded to each well. After 24 h, complete medium was removed and cells were treated with 200 µL bioconjugate solution in serum-free medium (concentration range 0.0032–50 µM, control wells were treated with serum-free medium). Medium was taken out after 6 h treatment (PTX conjugates) or 24 h (Dau conjugates), replaced by complete medium and incubation was continued. To determine the cell viability, medium was removed after 72 h and 100 µL resazurin solution [45] (10% Tox-8 in FBS-free medium) was added to each well, followed by additional incubation of 2–3 h. A Tecan infinite 200 pro microplate reader (Tecan Group Ltd., Zürich, Switzerland) was used for fluorescence detection (λ_Ex_ = 560 and λ_Em_ = 590 nm). Experiments were performed at least twice, using three parallels per concentration. The cell viability (and IC_50_ values) was calculated with GraphPad Prism using a nonlinear regression (sigmoidal dose-response).

### 4.3. Degradation of Drug Conjugates in Presence of Rat Liver Lysosomal Homogenate

The preparation of the rat liver lysosomal homogenate was carried out as recently reported [28]. A Qubit Protein Assay Kit (Thermo Fischer Scientific, Waltham, MA, USA) was used to examine the concentration of the lysosomal proteins.

Initially, a stock solution of the GnRH-III-Dau conjugates in water (5 µg/µL) was prepared. For the degradation reaction, the conjugates and the lysosomal homogenate were diluted together in 0.2 M NaOAc buffer (pH 5) to a concentration of 0.25 µg/µL each. These mixtures were kept at 37 °C and at distinct time points (5 min, 1 h, 2 h, 4 h, 8 h and 24 h), 15 µL of each sample were quenched with 2 µL acetic acid and analyzed by LC-MS (System I).

The GnRH-III-PTX conjugates were dissolved in DMSO to a concentration of 10 mM. The reaction was carried out in 0.2 M NaOAc buffer (pH 5) with 10 µM conjugate and 0.025 µg/mL lysosomal homogenate. The reaction mixtures were incubated at 37 °C and aliquots of 60 μL were taken at 5 min, 1 h, 2 h, 4 h, 8 h and 24 h and quenched with 5 μL of acetic acid. The analysis of the samples was performed by LC–MS (System-II).

### 4.4. Western Blotting

For the determination of the GnRH-receptor expression, Western blot analysis was performed with whole cell lysate. For each cell line, 10^6^ cells/well were seeded using six-well plates. The next day, PBS was used to wash the cells twice before cell lysis. The cells were incubated for 30 min on ice in 250 µL lysis buffer (50 mM Tris pH 7.4, 150 mM NaCl, 2 mM EDTA, 1% Triton-X and Protease Inhibitor Cocktail (Halt)). Afterwards, the samples were centrifuged at 20000× *g*. The protein concentration of the supernatant was examined with a Qubit Protein Assay Kit. The cell lysates were separated on a 10% Tris-tricine gel and then blotted to PVDF membrane (Merck Millipore). The GnRH-Rs were detected by an anti-GnRHR antibody (Proteintech, Rosemant, IL, USA, Catalog Number:19950-1AP, produced in rabbit, 1:1000) and incubation with an anti-rabbit-HRP secondary antibody (Santa Cruz Biotechnology, Dallas, TX, USA, produced in goat, 1:3000). ECL Substrate (Western Lightning Plus-ECL, PerkinElmer, Waltham, MA, USA) was used for detection of the Chemiluminescence. Then, an anti-actin primary antibody (Santa Cruz Biotechnology, produced in goat, 1:2000) was applied (after stripping of the membrane) and an anti-goat-HRP secondary antibody (Santa Cruz Biotechnology, produced in mouse, 1:3000) was used to detect actin (loading control).

### 4.5. Radioligand Binding Studies

A ligand competition assay with [^125^I]-GnRH-I-[^6^D-Trp] as radioligand was performed to examine the binding affinity of the GnRH-III-drug conjugates to GnRH-Rs expressed on human pituitary and human prostate cancer cells [10,11,38,39]. Human prostate cancer cells were derived from patients at the time of initial surgical treatment, while tissue samples from normal human pituitary (anterior lobe) were obtained by autopsy. All subjects were informed and gave their consent for inclusion before they participated in the study. The preparation of the cell membranes was carried out according to the literature [10,38,39,46]. The chloramines-T method was used to prepare radioiodinated GnRH-I agonist triptorelin which was purified by RP-HPLC [10,38,39,47]. The displacement of [^125^I]-GnRH-I-[^6^D-Trp] was studied in an in vitro ligand competition assay to determine the binding affinities of the non-radio-labeled GnRH-III bioconjugates to GnRH-RI [10,11,38,39]. Membrane homogenates containing 50–160 mg protein were incubated in duplicate or triplicate with 60–80,000 cpm [^125^I]-GnRH-I-[^6^D-Trp] and rising concentration (1 pM—1 µM) of nonradioactive bioconjugates as competitors in a total volume of 150 mL binding buffer. Protein concentration was determined by the method of Bradford using a Bio-Rad protein assay kit (Bio-Rad Laboratories, Hercules, CA, USA). The LIGAND-PC computerized curve-fitting program of Munson and Rodbard was used to determine the receptor binding characteristics and IC_50_ values [10,11,38,39].

## Data Availability

Not applicable.

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
