# Peer review of "Development and Biochemical Characterization of Self-Immolative Linker Containing GnRH-III-Drug Conjugates"

_ijms, 2022, doi:10.3390/ijms23095071_

Round 1

Reviewer 1 Report

The manuscript intitled ‘Development and biochemical characterization of self-immolative linker containing GnRH-III-drug conjugates’ authors have synthesized and evaluated all the GnRH-III conjugates for their anticancer activity. Over all paper is interesting and can be accepted after minor modification. Below are some minor comments for the authors need to be incorporated before publication.

  1. In the discussion part authors should incorporate the NMR and mass spectrometry discussion of all the intermediate and the target compounds formation, which is lacking in the manuscript.
  2. Authors should assign the characteristic peaks in the NMR data provided in the supplementary material.
  3. NMR spectra of the target compounds are missing in the supplementary material, authors should provide all the spectra in the supplementary file.

Author Response

Responses to the comments of Reviewer 1

  1. In the discussion part authors should incorporate the NMR and mass spectrometry discussion of all the intermediate and the target compounds formation, which is lacking in the manuscript

Response: Thank you for the comment. We adjusted the manuscript accordingly and added a short paragraph about the synthesis and analysis of the compounds to the discussion (line 327-334). Apart from that we added some more detail in the results section to provide more information (line 141-163). The formation of the products (the final linkers, the corresponding intermediates and the final conjugates) was either confirmed by NMR or ESI-MS. Considering that the main parts of the synthesis route are published several times we decided to keep the discussion of the synthesis concise [1–10].

  1. Authors should assign the characteristic peaks in the NMR data provided in the supplementary material.

Response: We agree with the Reviewer and assigned the characteristic peaks of the NMR data in the supplementary part.

  1. NMR spectra of the target compounds are missing in the supplementary material, authors should provide all the spectra in the supplementary file.

Response: We included the corresponding NMR spectra of the stated NMR data in the manuscript. The final GnRH-III-drug conjugates were solely identified and confirmed by analytical HPLC and ESI-MS. Considering that the synthetic route of the dipeptidyl-PAB-drug linker is well established we did not intent to provide NMR characterization of the peptide conjugates next to the HPLC and ESI-MS analysis.

References

  1. Hochdörffer, K.; Abu Ajaj, K.; Schäfer-Obodozie, C.; Kratz, F. Development of Novel Bisphosphonate Prodrugs of Doxorubicin for Targeting Bone Metastases That Are Cleaved PH Dependently or by Cathepsin B: Synthesis, Cleavage Properties, and Binding Properties to Hydroxyapatite As Well As Bone Matrix. J. Med. Chem. 2012, 55, 7502–7515, doi:10.1021/jm300493m.
  2. Dal Corso, A.; Caruso, M.; Belvisi, L.; Arosio, D.; Piarulli, U.; Albanese, C.; Gasparri, F.; Marsiglio, A.; Sola, F.; Troiani, S.; et al. Synthesis and Biological Evaluation of RGD Peptidomimetic–Paclitaxel Conjugates Bearing Lysosomally Cleavable Linkers. Chemistry – A European Journal 2015, 21, 6921–6929, doi:10.1002/chem.201500158.
  3. Dubowchik, G.M.; Firestone, R.A.; Padilla, L.; Willner, D.; Hofstead, S.J.; Mosure, K.; Knipe, J.O.; Lasch, S.J.; Trail, P.A. Cathepsin B-Labile Dipeptide Linkers for Lysosomal Release of Doxorubicin from Internalizing Immunoconjugates: Model Studies of Enzymatic Drug Release and Antigen-Specific in Vitro Anticancer Activity. Bioconjug. Chem. 2002, 13, 855–869, doi:10.1021/bc025536j.
  4. Owen, D.; Tsegay, S.; Shengule, S.; Reitano, P.; Porter, C.; Johnston, A.; Yuen, D. Targeted Dendrimer Conjugates 2021. Patent: WO2021035310A1
  5. Kratz, F.; Hochdoerffer, K. Bisphosphonate-Prodrugs 2012. Patent: AU2012219936A1
  6. Bouchard, H.; Brun, M.-P.; Hubert, P. Peptidic Linkers and Cryptophycin Conjugates, Useful in Therapy, and Their Preparation 2018. Patent: WO2012113571A1
  7. Lange, J.; Anderson, R.J.; Marshall, A.J.; Chan, S.T.S.; Bilbrough, T.S.; Gasser, O.; Gonzalez-Lopez, C.; Salio, M.; Cerundolo, V.; Hermans, I.F.; et al. The Chemical Synthesis, Stability, and Activity of MAIT Cell Prodrug Agonists That Access MR1 in Recycling Endosomes. ACS Chem. Biol. 2020, 15, 437–445, doi:10.1021/acschembio.9b00902.
  8. Senter, P.D.; Doronina, S.; Toki, B.E. Drug Conjugates and Their Use for Treating Cancer, an Autoimmune Disease or an Infectious Disease 2004. Patent: WO2004010957A2
  9. Borbély, A.; Figueras, E.; Martins, A.; Bodero, L.; Raposo Moreira Dias, A.; López Rivas, P.; Pina, A.; Arosio, D.; Gallinari, P.; Frese, M.; et al. Conjugates of Cryptophycin and RGD or IsoDGR Peptidomimetics for Targeted Drug Delivery. ChemistryOpen 2019, 8, 737–742, doi:10.1002/open.201900110.
  10. Borbély, A.; Figueras, E.; Martins, A.; Esposito, S.; Auciello, G.; Monteagudo, E.; Di Marco, A.; Summa, V.; Cordella, P.; Perego, R.; et al. Synthesis and Biological Evaluation of RGD–Cryptophycin Conjugates for Targeted Drug Delivery. Pharmaceutics 2019, 11, 151, doi:10.3390/pharmaceutics11040151.

Reviewer 2 Report

General comments:

This study underlines the high value of GnRH-III-based homing devices and the application of Cathepsin B cleavable linker systems for the development of small molecule drug conjugates (SMDCs).

Major comments:

A positive control such as the clinical drugs (cisplatin or others) for cancer cell cytotoxicity is suggested.

Minor comments:

  1. Table 1 & 2: n = ?. Please mention the replicate number
  2. Is there any reference using resazurin for viability assay? Please provide it. What is the principle for it?

Author Response

Responses to the comments of Reviewer 2

Major comments:

A positive control such as the clinical drugs (cisplatin or others) for cancer cell cytotoxicity is suggested.

Response: Thank you for the comment. We agree with the Reviewer that it is advisable to include a clinical drug as positive control in cytotoxicity studies. Therefore, we tested not only the developed GnRH-III-drug conjugates, but also the free anticancer drugs daunorubicin and paclitaxel. Paclitaxel belongs to the first-line chemotherapy drugs for pancreatic, ovarian, endometrial cancers and other malignancies [1]. Moreover, it has been shown that PTX in combination with platin based chemotherapeutics (cisplatin or carboplatin) was successfully used as treatment regime in advanced ovarian cancer [2,3]. Due to this, PTX can be considered as clinical drug-based positive control. From our point of view, the usage of PTX as positive control in our study is more informative than that of cisplatin, since it is helpful to interpret and understand the results of the GnRH-III-PTX conjugates. Considering that the development of cisplatin-resistance in ovarian tumors as well as acquired cisplatin resistance in pancreatic cancer cells has been reported [4–6], the application of cisplatin as positive control next to PTX might provide only limited information. Nevertheless, we go along with the Reviewers point of view that the usage of an additional clinical drug as positive control which circumvent drug resistance mechanism might be favorable.

Minor comments:

  1. Table 1 & 2: n = ?. Please mention the replicate number

Response: We thank the Reviewer for the advice. The replication number was added in Table 1 and 2

  1. Is there any reference using resazurin for viability assay? Please provide it. What is the principle for it?

Response: Yes, there are many publications confirming the use of resazurin for cell viability assay and we added a citation of Page et al. (1993) to our manuscript. In general, more than 1800 publications are listed at CAS SciFindern (accessed on April, 22st 2022) when using “resazurin cell viability assay” for query. The three most relevant (search: substances, sort by relevance) were together cited more than 3500 times (2160, 1021 and 423 times respectively) [7–10]. According to the active metabolism of viable cells resazurin is reduced by NADH to the resorufin product (Figure A [11]) which is pink and highly fluorescent (λEx = 560 and λEm = 590 nm). The quantity of resorufin produced is proportional to the number of viable cells [12]. Apart from that resazurin-based assay kits as well as resazurin powder are commercially available by different suppliers [12] e.g.

  • CellTiter-Blue® Cell Viability Assay (Promega Corporation Cat.# G8081)
  • In Vitro Toxicology Assay Kit, Resazurin based (Sigma-Aldrich Cat.# TOX8-1KT)
  • alamarBlue®—Rapid & Accurate Cell Health Indicator (Life Technologies, Inc. Cat.# DAL1100)
  • alamarBlue® (AbD Serotech Cat.# BUF012B)
  • Resazurin sodium salt. (Sigma-Aldrich Cat.# R7017-1G)

Figure A. The principle of resazurin cell viability assay [https://www.creative-bioarray.com/support/resazurin-cell-viability-assay.htm (accessed on April, 21st 2022)]

References

  1. Guo, F.; Li, J.; Qi, Y.; Hou, J.; Chen, H.; Jiang, S.-W. HE4 Overexpression Decreases Pancreatic Cancer Capan-1 Cell Sensitivity to Paclitaxel via Cell Cycle Regulation. Cancer Cell International 2020, 20, 163, doi:10.1186/s12935-020-01248-1.
  2. Armstrong, D.K.; Bundy, B.; Wenzel, L.; Huang, H.Q.; Baergen, R.; Lele, S.; Copeland, L.J.; Walker, J.L.; Burger, R.A. Intraperitoneal Cisplatin and Paclitaxel in Ovarian Cancer. New England Journal of Medicine 2006, 354, 34–43, doi:10.1056/NEJMoa052985.
  3. Kampan, N.C.; Madondo, M.T.; McNally, O.M.; Quinn, M.; Plebanski, M. Paclitaxel and Its Evolving Role in the Management of Ovarian Cancer. BioMed Research International 2015, 2015, e413076, doi:10.1155/2015/413076.
  4. Shen, D.-W.; Pouliot, L.M.; Hall, M.D.; Gottesman, M.M. Cisplatin Resistance: A Cellular Self-Defense Mechanism Resulting from Multiple Epigenetic and Genetic Changes. Pharmacol Rev 2012, 64, 706–721, doi:10.1124/pr.111.005637.
  5. Mezencev, R.; Matyunina, L.V.; Wagner, G.T.; McDonald, J.F. Acquired Resistance of Pancreatic Cancer Cells to Cisplatin Is Multifactorial with Cell Context-Dependent Involvement of Resistance Genes. Cancer Gene Ther 2016, 23, 446–453, doi:10.1038/cgt.2016.71.
  6. Miyata, Y.; Matsuo, T.; Nakamura, Y.; Yasuda, T.; Ohba, K.; Takehara, K.; Sakai, H. Expression of Class III Beta-Tubulin Predicts Prognosis in Patients with Cisplatin-Resistant Bladder Cancer Receiving Paclitaxel-Based Second-Line Chemotherapy. Anticancer Res 2018, 38, 1629–1635.
  7. 550-82-3 Reference Search | CAS SciFindern Available online: https://scifinder-n.cas.org/search/reference/62627a517966600b903bbb35/1 (accessed on 22 April 2022).
  8. O’Brien, J.; Wilson, I.; Orton, T.; Pognan, F. Investigation of the Alamar Blue (Resazurin) Fluorescent Dye for the Assessment of Mammalian Cell Cytotoxicity. Eur J Biochem 2000, 267, 5421–5426, doi:10.1046/j.1432-1327.2000.01606.x.
  9. Ansar Ahmed, S.; Gogal, R.M.; Walsh, J.E. A New Rapid and Simple Non-Radioactive Assay to Monitor and Determine the Proliferation of Lymphocytes: An Alternative to [3H]Thymidine Incorporation Assay. Journal of Immunological Methods 1994, 170, 211–224, doi:10.1016/0022-1759(94)90396-4.
  10. Page, B.; Page, M.; Noel, C. A NEW FLUOROMETRIC ASSAY FOR CYTOTOXICITY MEASUREMENTS IN-VITRO. International Journal of Oncology 1993, 3, 473–476, doi:10.3892/ijo.3.3.473.
  11. Resazurin Cell Viability Assay | Creative Bioarray Available online: https://www.creative-bioarray.com/support/resazurin-cell-viability-assay.htm (accessed on 23 April 2022).
  12. Riss, T.L.; Moravec, R.A.; Niles, A.L.; Duellman, S.; Benink, H.A.; Worzella, T.J.; Minor, L. Cell Viability Assays; Eli Lilly & Company and the National Center for Advancing Translational Sciences, 2016;

Reviewer 3 Report

Generally, well written explained, and easy to read the manuscript. Possibly better with the following:

1) An abstract figure-ish summary scheme/figure to wrap up the investigation in the discussion section. 

2) Could be quite useful for many readers: summarizing lysosomal enzymes of interest and how the current designs fit and compared to others. 

Author Response

Responses to the comments of Reviewer 3

Generally, well written explained, and easy to read the manuscript. Possibly better with the following:

  1. An abstract figure-ish summary scheme/figure to wrap up the investigation in the discussion section.

Response: We thank the Reviewer for the advice. We prepared such a summarizing scheme to visualize the investigations in the discussion section (page 13).

  1. Could be quite useful for many readers: summarizing lysosomal enzymes of interest and how the current designs fit and compared to others.

Response: Thanks a lot for your comment and raising up this topic. We added some more information the discussion part (line 366-378) and referred to one of our previous publications, where we used another linker design and discussed in more detail the role of different cathepsins on the release of active drug metabolites. Moreover, a short comparison of the current linker design and the previous studied oxime-bond linked GnRH-III-Dau conjugates was included in the manuscript.